# Exploring the Potential of *Myrothamnus flabellifolius* Welw. (Resurrection Tree) as a Phytogenic Feed Additive in Animal Nutrition

**DOI:** 10.3390/ani12151973

**Published:** 2022-08-03

**Authors:** Carlos Wyson Tawanda Nantapo, Upenyu Marume

**Affiliations:** 1Department of Animal Sciences, School of Agricultural Sciences, Faculty of Agriculture, Science and Technology, North-West University, P Bag X 2046, Mmabatho 2735, South Africa; nantapocarlos@gmail.com; 2Food Security and Safety Niche Area, Faculty of Agriculture, Science and Technology, North-West University, P Bag X 2046, Mmabatho 2735, South Africa

**Keywords:** antimicrobial, antioxidants, bioaccessibility, bioavailability, essential oil, gut function, phytogenic additive, microencapsulation

## Abstract

**Simple Summary:**

The unregulated use of in-feed antibiotic growth promoters has received widespread condemnation due to an increase in cases of antibiotic-resistant microbes. This has fueled an ever-growing demand for new sources of natural and safe alternative products with minimal impacts on the environment and human health in animal production. *Myrothamnus flabellifolius*, as a phytogenic feed additive, fits this description, as it is a natural plant containing high amounts of secondary metabolites necessary for cell function, regulation, and protection for improved animal growth, performance, and health. With some limitations towards its use, several processing and combination strategies are available to unlock nutrients and explore its potential in animal production, as described in this review.

**Abstract:**

*Myrothamnus flabellifolius* (Welw.) is used in African traditional medicine for the treatment of depression and mental disorder, asthma, infectious diseases, respiratory, inflammation, epilepsy, heart, wound, backaches, diabetes, kidney ailments, hypertension, hemorrhoids, gingivitis, shingles, stroke, and skins conditions. The effectiveness of *M. flabellifolius* is due to the presence of several secondary metabolites that have demonstrated efficacy in other cell and animal models. These metabolites are key in cell regulation and function and have potential use in animal production due to antimicrobial and antioxidant properties, for an improvement in growth performance, feed quality and palatability, gut microbial environment, function, and animal health. The purpose of this review is to provide a detailed account on the potential use of *M. flabellifolius* in animal nutrition. Limitations towards the use of this plant in animal nutrition, including toxicity, economic, and financial issues are discussed. Finally, novel strategies and technologies, e.g., microencapsulation, microbial fermentation, and essential oil extraction, used to unlock and improve nutrient bioaccessibility and bioavailability are clearly discussed towards the potential use of *M. flabellifolius* as a phytogenic additive in animal diets.

## 1. Introduction

Mufandichimuka (Shona), Umfavuke (Ndebele), Uvukabafile (Zulu), or Moritela Tshwene (Setswana) are vernacular names that reflect the remarkable and unique characteristics of the angiosperm, desiccation-tolerant woody-shrub *Myrothamnus flabellifolius* (Welw.) (Family: Myrothamnaceae), commonly referred to as the Resurrection tree/bush. The plant grows on shallow, well-drained crevices in sandstone and granite outcrops, surviving dehydration to a near-dry quiescent state for a year or more. The plant is able to resurrect or hydrate and revitalize metabolic, photosynthetic activity and growth 24–72 h after the first substantial amount of summer rainfall [1,2]. In the air-dried state, the plant can lose up to 95% of its relative cellular water content [3,4] and approximately 30–50% chlorophyll content, depending on the severity of the dry spell [5]. This remarkable feature of its ‘resurrection’ has been noted as a symbol of hope in many African communities’ tradition/folklore medicine. Its decoction or tea infusion has been used to ‘bring sufferers back to life’ or endow patients with the plant’s unusual ability to come back to life, especially in cases of depression and mental disorders [6,7]. The medicinal uses also extend to treatment of asthma, infectious diseases, respiratory, inflammation, epilepsy, heart, wound, backaches, diabetes, kidney ailments, hypertension, hemorrhoids, gingivitis, shingles, stroke, and skins conditions [8,9,10,11]. The broad range of benefits from the resurrection tree have been demonstrated widely in cell cultures, rodent models, and some human test targets. These observed benefits are linked to a very large and diverse pool of secondary plant metabolites and active ingredients, which modulate physiological processes in animal cells [12]. The actual quality and quantity of active ingredients in this plant’s species and varieties are yet to be fully understood. In addition, the mode of action has not yet been fully postulated due to huge numbers and vast differences in ingredient composition and methods used in many studies [13]. However, the presence of active ingredients pinocarvone, trans-pinocarveol, limonene, trans-p-menth-1-(7)-8-diene-2-ol, cis-p-menth-1-(7)-8-diene-2-ol [8], sugars; fructose, saccharose, glucose, α-α-trehalose, raffinose, stachiose [14], galloylquinic acids, and derivatives of galloyl glucose hexahydroxydiphenic acid [15], which are linked to several pharmacological properties, including antioxidant, antimicrobial, antiviral, antidiabetic, and anticancer, has been detected in *M. flabellifolius* [7].

The presence of active ingredients with important pharmacological properties plus increasing evidence of its exceptional benefits suggests its potential use in ethnoveterinary and animal nutrition as an additive for growth improvement, gut environment, and animal health in several animal species. The potential use of *M. flabellifolius* could be timely, considering the ever-growing demand by humans for natural and safe products with minimum impact on the environment. There has been intensified attention on this woody shrub for commercialisation as functional tea, cosmetic products, and other pharmacological products. In contrast, little interest and scientific research has been devoted to elucidating the effects of the resurrection tree on many aspects of farm animal growth and performance. *M. flabellifolius* demonstrates great promise for use as a natural, safe, and sustainable feed additive and could be a plausible alternative to feed antibiotics in animal nutrition.

Local and global statistics predict a growing demand for natural products.

This review aims to coherently collate the fragmented data and comprehensively review the potential benefits of its use in animal nutrition, outlining the limitations and opportunities for its improvement.

## 2. Nutritional Composition of *Myrothamnus flabellifolius* (Welw.)

Nutritional components found in leaf extracts of *M. flabellifolius* include substantial amounts of dry matter, protein, fat, carbohydrates, dietary fibers, minerals and vitamins, fatty acids, reducing sugars, and cholesterol (3 mg/100 g) [16]. The same authors identified 18 essential and nonessential amino acids together with a high concentration of unsaturated fatty acids, including three, six, and nine fatty acids, suggesting potential nutritional benefits to monogastric animals.

The unusually high content and composition of sugars, including raffinose (2.49 g/100 g) and stachyose (2.18 g/100 g), in the leaves of *M. flabellifolius* play an important role in mechanisms involved in alleviating and adapting the plants to osmotic stress. The concentrations of these sugars varied according to moisture content in plants, with fully hydrated plants containing more sugars than desiccated plants. The same researchers [16] found raffinose and stachyose at 2.49 and 2.18 g/100 g, respectively, in *M. flabellifolius* tea extract. In addition to α-α-trehalose (3.3% *w*/*w*), [17] also identified raffinose (0.2% *w*/*w*) and stachyose (0.2% *w*/*w*), sucrose, saccharose, fructose, and fructane.

Trehalose (α-D-glucopyranosy-1,1-α-D-glucopyranoside) shortened α-α-trehalose is a very rare, extraordinarily stable, non-toxic, non-reducing disaccharide of glucose. It is found in a range of organisms, including lower plants, algae bacteria, insects, yeast, and invertebrates [18]. Trehalose is an important storage sugar responsible for cell osmotic stress protection. It is responsible for stabilising proteins and other cytoplasmic macromolecules and protecting them against reactive oxygen species [19]. Specifically, trehalose stabilises the proteins and lipids in the cellular bilayer by concentrating any remaining water close to proteins during dehydration. This helps to maintain their quaternary structure, a property that could be useful during water shortages in animal production systems in the world [20,21]. The natural functions, protective properties, technical qualities, and mechanisms of action of trehalose offer a wide range of industrial and practical opportunities for the feed, food, and biomedical industries. These include stable processing and extended shelf life of stored feeds, vaccines, and biomaterials, as well as the preservation and enhancement of organoleptic quality in fresh, dried, or frozen food, and feed ingredients without damage and loss of important biochemical components, including very labile enzymes and vitamins by offering desiccation tolerance [21]. As alluded to by Colaco and Roser [22], trehalose prevents the loss of aromatic volatiles, protects against denaturation, and reduces the effects of Maillard reactions during processing and accumulation of toxic by-products during storage and product stability, thus, maintaining a higher nutritional content in feeds. There is a dearth of information on trehalose in animal nutrition and little is understood on how animal cells take up dietary trehalose. A few studies, including Jun et al. [23], noted that unlike other disaccharides, dietary trehalose is directly absorbed into the blood stream of broilers without being broken down into glucose units, through simple diffusion in chicken brush border membrane of enterocytes [24]. The quantities of trehalose found in *M. flabellifolius*, in many studies, hydrated 26.0 ± 7.7 mgg^−1^ vs. desiccated 34.7 ± 4.3 mmg^−1^ dry weight [25], seem sufficient to illicit some reaction in vivo. In ruminant animals, the authors of [26,27] reported an improvement in oxidative status in blood, milk, and rumen fluid in cows supplemented with trehalose, whilst [28] reported on reduced medication frequency of pre-weaned calves and growth inhibition of pathogenic bacteria when trehalose was supplemented on milk replacers in pre-weaned calves. Chen et al. [29] highlighted the potential of trehalose as a prebiotic, which encourages growth in the host’s beneficial intrinsic probiotic bacteria. As such, Bhuiyan et al. [30] noted a weight increase in birds on higher trehalose but short-term supplementation. This was also corroborated by Iji et al. [31], who observed no feed efficiency differences but 7% greater mean body weights in broiler chicks with supplementation of trehalose at 20 g/kg. Kikusato et al. [32] observed improved body gains and reduced mRNA levels in innate immune response genes that function as inflammatory and anti-inflammatory cytokines, antigen receptors, and pro-oxidant enzymes. In that study, the authors supplemented diets with 0.5% (*w*/*w*) and observed comparable growth and suggested beneficial effects of trehalose on intestinal innate immunity. In contrast, in a follow-up study, Ruangpanit et al. [24] did not observe improved weight gains by supplementing trehalose to basal diets at 0.25, 0.50, and 0.75% but better villus surface area and villus height: crypt depth ratio at day 19 post hatch in broilers. The two studies concluded that trehalose improves the intestinal innate immune system by incorporating into epithelial cells to modulate cell renewal by autophagy in young chicks, which, in turn, directs nutrients from supporting immunity to growth improvement. The safety of trehalose in monogastric animals was explained by the presence of the gastric enzyme trehalase, which easily breaks down trehalose into simple glucose molecules, rendering it completely non-toxic and available for use by the animal, thus, preventing diarrhea in both chicks and mature birds. Further, Lee et al. [33] noted other benefits, including a protective effect against neurodegenerative disorders, protein aggregation, anti-apoptotic function, and improvement in cognitive function and mood in therapeutic models of young animals.

Researchers Bianchi et al. [34] found substantial amounts of glucosylglycerol (-glucopyranosyl-β-(1-≥)-glycerol), α-α-trehalose, and arbutin, at 14, 19, and 25%, respectively, in water extracts of *M. flabelifollius*. Glucosylglycerol (GG) is an organic compound with osmolyte potential and prebiotic properties, reflected through microbial growth stimulation, gastrointestinal tolerance, pathogenic inhibition, and protecting the cell and protein from high salt concentration and other extreme conditions [35,36]. This is achieved by adjusting cellular osmotic potential of cells to levels that allow water uptake [37]. Wolf et al. [38] noted GG’s potential as an anti-aging proponent in pharmaceuticals and cosmetics industries. In broiler production, high-salinity levels from the domestic water supply interfere with cellular function. The ability of cells to adapt to changes in external osmolarity can be enhanced by the inclusion of compatible or osmo-protective solutes that do not interfere with cell metabolism and help prevent denaturation of macromolecules [39]. This can be necessitated by dietary inclusion of GG-rich *M. flabelifollius*. In addition, this compound is effective in decreasing caloric intake by suppressing the intestinal disaccharide action, suppression of hepatic glucose activity, thus, limiting mechanisms associated with obesity with the finisher phase of broiler production [40,41].

Functional oligosaccharides (α-galactosides), raffinose, and stachyose have long been considered antinutritional factors in poultry as they are indigestible due to a lack of α-galactosidase digestive enzymes [42]. However, recent studies have highlighted their benefits, including their ability in promoting animal and gut health and microbiota. Qiang et al. [43] noted the effects of raffinose and stachyose in promoting a healthy balance of intestinal microflora by increasing salutary bifidobacteria and lactobacilli, reducing gastrointestinal infections and suppressing diarrhoea. In addition, they noted the noncariogenic properties of dietary oligosaccharides and their ability to supply the smallest amount of energy amongst disaccharides, as well as their inability to increase blood glucose levels, hence, reducing the chances of obesity in animals. This was corroborated by Patel and Goyal [44], who found a positive modulation of cecal microflora in the gut of broilers supplanted with oligosaccharides, immunity, and improved antioxidant enzymes. Administration of raffinose and stachyose as an in-water supplemented prebiotic, at levels equivalent to those present in water extracted *M. flabellifolius,* increased the numbers of beneficial lactobacilli and bifidobacterial in chicken faeces [45]. Small intestine morphometric and microstructure parameters were improved by diet supplementation with pure forms of raffinose family oligosaccharides [46]. This was also corroborated by Xu et al. [47], who saw a marked increase in growth performance and improved intestinal morphology and microbial community in Sturgeon fed diets supplemented with raffinose. These findings led to the amplified global manufacture of raffinose and stachyose synthetics for use in animal nutrition [31]. However, restraint must be used in choosing the dose rate as levels of 4.0, 8.0, 12.0, and 16.0 g/kg of stachyose in broiler chicken basal diets led to depressed body weights, nutrient digestibility, and no positive effect on gut pH and microflora [48]. Others found an increase in growth performance at 0.5% supplemented stachyose but reduced performance at any higher dose. This suggested a dose-dependent relationship between dietary supplements and growth performance, which may require cytotoxicity and genotoxicity analysis prior to feeding. Therefore, these studies confirm that the amounts of natural α-galactosides, raffinose, and stachyose present in *M. flabellifolius* are enough to illicit positive changes when it is included in the diet and may act as an alternative to antibiotics or prebiotics in animal nutrition.

Arbutin (hydroquinone-β-D-glucopyronoside) and its derivatives, such as 2,3-di-O-galloylarbutin, are naturally occurring glucopyranosides found in many plant species. It is known to have antiparasitic characteristics that deter both fungal and herbivore attack, as well as antioxidant, anti-inflammatory, and antitumour properties in vitro and in vivo [49,50,51]. This compound offers control and protection against cell damage caused by oxidative injury stress [52] and cell viability restoration, a useful property for protection of the intestinal cell wall lining in growing chicks. The presence of arbutin in an in vitro consecutive batch culture technique on rumen activity resulted in improved gas and VFA production and overall improved rumen microbial activity compared to other tested phenolic compounds [53]. With respect to *M. flabelifollius*, the negative digestive effects associated with the antinutritional properties of some compounds present will be countered by the positive effects of simpler sugar molecules released in the later stages of the gastrointestinal tract. However, ethanol extracts containing aglycone derivative and active metabolite hydroquinone have been found to cause detrimental effects, such as DNA and chromosomal damage, cell damage, and mutagenic activity, at certain doses [54]. Therefore, further studies are needed to fully investigate the appropriate dose, extraction, or chemical and biotechnological synthesis methods for derivatives that are environmentally and animal friendly [55].

## 3. Phytochemical Composition of *Myrothamnus flabellifolius* (Welw.)

The bioactive, non-nutritive compounds or secondary metabolites present in *M. flabellifolius* (Welw.) are grouped according to their chemical structure. Several classes of secondary metabolites have been reported by several authors, including polyphenols, phytosterols, terpenoids, and glucosinolates. These phytochemicals have both lethal and beneficial effects on the physiological functioning (biological activity) of the cell and body, but these can only be fully understood through knowledge of their individual structure [56].

Most secondary metabolites in the resurrection plant are synthesised to protect and defend against damage and disease. The secondary metabolites’ nature and action, both in vivo and in vitro, suggest potential in veterinary or biomedical applications and drug development. However, there is little knowledge on the specific action of each metabolite on a specific physiological response. Additionally, the reported biological activities are from experiments involving crude extracts and in combination with other extracts [19]. Several compounds present in the essential oil of *M. flabellifolius* have been reported. Cheikhyoussef et al. [56] reported the presence of 12 classes of phytochemicals, including flavonoids, anthocyanins, alkaloids, steroids, terpenoids, triterpenes, cardiac glucosides, saponins, phlobatanins, tannins, and polyphenols, through Soxhlet ethanol–methanol–water solvent extraction of leaves and twigs, as depicted in Table 1.

Polyphenols are the most diverse group of more than 8000 plant-synthesised phytochemicals identified, which, esterified, exist with monosaccharides or polysaccharides (glycosides) or free aglycones. The compounds are produced by plants to protect against tissue damage, stress, and diseases, and a wide range of other biological properties [61]. Gessner et al. [12] divided plant polyphenols into flavonoid groups, which share two benzene rings connected by three carbon atoms, forming an oxygenated heterocycle and non-flavonoids containing an aromatic ring with one or more hydroxyl. These structures and galloyl moieties influence the chemical and biological property of each polyphenolic compound. Moore et al. [1] identified 3,4,5 tri-O-galloylquicic acid as the main polyphenolic compound present in *M. flabellifolius*. In addition, smaller quantities of multiple galloylation products, such as gallic acid, its ellagic acid esters, and other higher-molecular-weight galloylquinic acids, were quantified in plants collected in Namibia and South Africa. They further demonstrated that these tannin compounds constituted 50% of the dry mass of hydrated leaves and a third of the dry mass of desiccated leaves [56]. This compound, when fractionated and screened using a novel “ethidium bromide”-based fluorescence assay, demonstrated high antiviral properties by inhibiting viral (M-MLV and HIV-1) reverse transcriptase enzymes in vitro. This is attributed to the tannin characteristic of 3,4,5 tri-O-galloylquinic acid, which interacts with cell membrane proteins and binds with the viral coat proteins [62]. This tannin characteristic is detrimental at high doses and may cause denaturing of digestive enzymes, low digestion and absorption of nutrients, and, thus, low productivity in animals. In addition, the same authors further explained some challenges with the use of 3,4,5 tri-O-galloylquinic in antiviral therapies. These include its affinity for competitive and nonspecific binding to other proteins, which reduces concentration in cells and, thus, a high-concentration supply is needed in the body to effectively inhibit the HIV-RT for improved efficacy. In contrast, the non-specificity and multiple viral target “sites” nature of 3,4,5 tri-O-galloylquinic and its related compounds may be viewed as a potential avenue for the development of an antiviral remedy that targets multiple monogastric animal gastrointestinal tract viruses, including rotaviruses, coronaviruses, enteroviruses, adenoviruses, astroviruses, and reoviruses. This is supported by a recent shift from traditional methods of single-target search to multiple-target multi-level comprehensive or Network pharmacology [63]. Other properties include reduced total lesion area, improved overall gastroprotective activity, and cytotoxicity against gastric adenocarcinoma cells of ulcer models in mice from methylated galloylquinic acid derivatives [64]. In a study by Lemos et al. [65], an increase in mucus production and decrease in oxidative stress in the gut could be related to the gastroprotective effects effected by the antioxidant properties of these secondary metabolites in plant extracts of *M. flabellifolius*. Furthermore, Moore et al. [2] noted the protective free radical scavenging ability of 3,4,5 tri-O-galloylquicic against induced oxidation on linoleic acid, an essential fatty acid in animal nutrition at low levels.

Quercetin (3,5,7,3′,4′-pentahydroxyflavone) and its derivatives, 3-O-β-d-galactoseides, -glucosides, -gluconorides, and 3-O-α-L-rhamnosides, are natural polyphenolic flavonoid secondary metabolites, with relatively higher bioavailability than other phytochemicals commonly found in fruits and vegetables. Recent literature has described the antihyperlipidemic and antihyperglycemic properties of quercetin [66,67]. These remedial physiological processes involve inhibiting 3-hydroxy-3-methyl-glutaryl (HMG)-CoA reductase and a decline in the activity of glucose-6-phosphate dehydrogenase in the liver, thus, reducing muscle and serum triglycerides and cholesterol accumulation. For the antihyperglycemic process, the activation of L-type calcium channels promotes insulin production in beta cells, leading to lower blood sugar [68]. An accumulation of excessive saturated fat, which is susceptible to oxidation, is a common but unwanted genetic and production disorder, with negative effects on the antioxidant system, resulting in poor meat quality and production losses [69]. Supplementing the diet with quercetin-dense sources in Arbor Acre broilers prevents and limits the damage exerted by oxidative stress [70]. The use of in-feed quercetin-rich additive phytogenic essential oils and extracts, such *M. flabellifolius,* to attenuate the effects of oxidative stress, modulate lipid peroxidation and protect unsaturated fatty acids, improve antioxidant status in broiler chickens, reduce blood and cellular total cholesterol and triglycerides, with a concomitant decrease in abdominal fat deposition, is a benefit to both farmers and consumers. Finally, [71] described the potential additive benefits of dietary quercetin, which suggests this product as a promising alternative multi-functional, consumer-preferred natural phytogenic growth promoter in broiler production.

Terpenoids and related compounds were previously reported to exhibit antiviral and bactericidal properties and used as pharmaceutical mixtures for the treatment of respiratory tract disorders. Pinocarvone and trans-pinocarveol are monoterpenes abundantly found in the essential oils of *M. flabellifolius*. Flavonols with a dihydroxylated B-ring, including quercetin-3-O-α-L-rhamnopyranoside (syn. Quercitrin), quercetin-3-O-β-D-glucopyranoside (syn. Isoquercitrin), and quercetin-3-O-β-D-galactopyranoside (syn. Hyperoside), were identified in *M. flabellifolius* [17]. In addition, kaempferol glycosides; kaempferol-3-O-β-D-glucopyranoside (syn. astragalin), aempferol-3-O-β-D-glucuronoopyranoside, empferol-3-O-β-D-galactopyranoside (syn. Trifolin), kaempferol-3-O-β-D-glucuronoopyranoside, and kaempferol-3-O-α-L-rhamnopyranoside (syn. Afzelin), were isolated and identified in smaller proportions in the same study. In another study, Rasoanaivo et al. [72] found higher trans-pinocarveol at 35.6–36.3% and pinocarvone at 19.8–20.0% in a species variety, *Myrothamnus moschatus* (Baillon) Niedenzu, endemic to Madagascar. The presence of 85 compounds, namely pinocarvone (11.13%), trans-pinocarveol (19.57%), limonene (6.09%), trans-p-menth-1-(7)-8-diene-2-ol (7.43% and cis-p-menth-1-(7)-8-diene-2-ol (6.76%), and others, representing 87.79% of the essential oil from hydro-distilled above-ground *M. flabellifolius* material, was reported [8]. The authors also assessed the cytotoxic potential of the essential oil, as well as antimicrobial, antifungal, and antioxidant potential. This was confirmed by Zorzetto et al. [73], who linked the presence of pinocarvone and related moieties in pharmaceutical stimulants and exploratory remedies used in the treatment of respiratory diseases. Further analysis revealed a link between the presence of trans-pinocarveol and stomachic and gastroprotective activity, enhanced digestive action and antimicrobial properties, displaying its potential in animal gut physiology improvement as a prebiotic in fusions in veterinary pharmacology [74].

## 4. Potential Animal Production and Health Benefits

The diversity of phytonutrients identified in *M. flabellifolius* makes it a significant potential source for a variety of beneficial ethnoveterinary uses. However, the phytochemical variability and rich diversity of phytocompounds from the different phylogenetic and metabolic signatures, sourced from different populations and locations, makes it difficult for standardisation of this medicinal plant and its derived products [3,4]. Hereafter is a summary of phytochemical properties observed in *M. flabellifolius,* which are further presented in Figure 1.

### 4.1. Antioxidant Effects

Fotina et al. [75] described the dietary plant polyphenolic compounds’ antioxidant protection and immunity-promoting properties, which help maintain a balance between the formation of oxidants and their detoxification and eradication, by aiding the antioxidant system along the main target site or gut lining [12]. Cheikhyoussef [56] and Bentley [3] described *M. flabellifolius* as a potent and valid natural source of antioxidants due to high quantities of the total phenolic content, especially in the Namibian variety. This was supported by observations of high total flavonoid content and total phenolic content ranges of 1.43 ± 0.03 to 3.49 ± 0.15 mg equivalents for quercetin and 372.42 ± 0.21 to 375.14 ± 0.21 mg gallic acid equivalent, respectively. As reported above, the presence of 3,4,5 tri-O-galloylquinic acid in an aqueous solution of linoleic acid resulted in a decreased rate of linoleic acid oxidation by approximately 95%. In addition, only a very low relative concentration was needed to effect antioxidant protection of unsaturated lipids. This suggests a concentration-dependent release and sequestration mechanism, a property necessary for in-feed additives. When added and mixed with animal feeds, 3,4,5 tri-O-galloylquinic acid and its derivatives reduce the rate of rancidity and build-up of toxic oxidation products, ensuring a longer shelf life and protected nutrient profile, especially of polyunsaturated fatty acids (PUFA), such as arachidonic acid (AA, 20:4n-6) and docosahexaenoic acid (DHA, 22:6n-3) [76]. Other studies by Bhebhe et al. [77], Ajao and Ashafa [58], Engelhardt et al. [17], and Chaturvedi et al. [11] confirm the high antioxidant capability using different assay methods and support its possible use in animal diets to target and improve gut health.

### 4.2. Antimicrobial Potential

The antimicrobial potential of test compounds can be measured by how they are able to slow down or destroy unwanted microbes. Minimum inhibitory or bactericidal concentration (MIC/MBC) is the lowest concentration of antimicrobial agent that completely inhibits the growth of the organisms or concentration, where 99.9% of microbes are killed. It is worth noting that essential oils have higher biological activity compared to the raw materials they were extracted from and have received widespread support as alternatives to antibiotic growth promoters, as they are able to reduce MIC and MBC towards multi-drug-resistant bacteria and are able to maintain an optimum balance of microflora diversity and population in the gastrointestinal tract (Eubiosis). The antimicrobial properties of plant-extracted phytochemicals in animal health and nutrition have been extensively researched, although there are still gaps in knowledge regarding their mechanisms of action towards different microbial populations, on target sites and within a feed matrix, and in the presence of other phytochemicals exerting different actions. However, others postulated the effects of the presence of a combination of active ingredients towards inhibition of the ATPase enzymes system responsible for energy generation, disruption of cell permeability, and cytoplasmic membrane, causing the release of intracellular compounds, leading to microbial cell death. However, some experimental work did not produce any positive effects and these discrepancies were related to management and sanitary conditions, environment, variability in animal scenarios, and phytochemicals [57]. Disc diffusion assays on hydro-distilled oils of *M. flabellifolius* showed antimicrobial effectiveness against pathogenic microbes, such as *Escherichia coli*, *Staphylococcus aureus*, *Enterococcus faecalis*, *Proteus vulgaris*, *Serratia odorifera*, *Salmonella typhimurium*, *Candida albicans*, *Cryptococcus neoformans*, *Alternaria alternata, Aspergillus niger,* and *Pseudomonas aeruginosa* [8]. This was linked to the presence of pinocarvone, trans-pinocarveol, limonene, trans-p-menth-1-(7)-8-diene-2-ol, and cis-p-menth-1-(7)-8-diene-2-ol. However, the same authors expressed the need for caution in correlating antimicrobial activity to the action of major essential oil compounds but tentatively to essential oil compounds through structure activity relationships. Bactericidal, candidacy, and fungicidal properties, good rapid death kinetic figures, and lower MIC values of *M. flabellifolius* against *S. aureus* in a time–kill study under 1 h were observed [78]. In addition, cidal effect properties against three pathogens, such *Candida albicans, Staphylococuss aureas, and Klebsiella pneumoniae,* of *M. flabellifolius* were discussed by [79]. The inclusion of *M. flabellifolius* will take a dual role in reducing pathogenic organisms in both the gut and in feed and improve feed preservation. However, it should be noted that to improve the efficacy by *M. flabellifolius* against several pathogenic microbes and reduce the advance of antimicrobial resistance of any feed formulation, the concept of antimicrobial synergy through combination with other compounds should be in principle, as outlined later in the review. Additionally, the cleansing of the digestive tract helps reduce ATP or energy losses during inflammation and immune responses [80], prevent diseases, promote improved feed and nutrient digestion, and absorption and assimilation for growth performance and production [76].

### 4.3. Growth Performance and Health Effects

Several compounds present in phytogenic feed additives are known to impart positive attributes on the feed taste and palatability, which, in turn, helps improve animal productivity. To our knowledge, no known studies have been carried out to test the effect of *M. flabellifolius* on feed quality, growth performance, and gut and immune function. We noted that several compounds present in this plant have been demonstrated to effect beneficial effects on gut function, such as spasmolytic, laxative, and against flatulence [81], stimulation of the digestive secretions, such as mucus, saliva, and bile, enzymes, such as trypsin, lipase, and amylase in broilers, reducing diarrhoea and appetite, and improving feed conversion efficiency, growth, and carcass characteristics [82]. As mentioned earlier, trehalose may help stabilise the feed proteins and lipids by concentrating any remaining water close to proteins during dehydration to maintain their quaternary structure, a property that could be useful during feed storage [21]. As a growth stimulator and replacement of antibiotic growth promoters, the flavonoid polyphenol constituents help in improving feed transit time, stabilise gut microflora, adjust to villi and crypts sizes and depths in the jejunum and colon, reducing intestinal disorders and pathogenic microbe burden. Meat and meat product shelf-life quality is extended through the use of several bioactive compounds present in *M. flabellifolius*.

## 5. Limitations to the Use of *M. flabellifolius* (Welw.) in Animal Nutrition

A few research studies have explored and elucidated the ethnopharmacological effects of *M. flabellifolius,* but none on its potential application in ethnoveterinary. Generally, results lack reliability and consistency due to variation in composition of individual ingredient purity and total compound profile in each individual herb or plant, which affects the total dosage that can be offered to animals and accuracy in quantifying dose–effect measurements. Described below are some of the constraints, which limit the potential use of *M. flabellifolius* in animal studies.

### 5.1. Low Biomass and Seasonal Availability

The plant is indigenous to Southern Africa and has been classified as underutilised and only harvested by herbalists or rural communities to treat many ailments. The plant is not a high-biomass producer and the low yields of above-ground biomass of *M. flabellifolius* occur only in the summer season, provided there is enough rain to stimulate resurrection. In Zimbabwe, informal collectors for private companies usually only manage to collect a maximum of 20–30 kg of twigs per day. The quantity collected per day increases with larger topography outlined with rocky outcrops, along cracks and crevices in rocks and higher yields in summer (September–May). The low-biomass production is, however, a significant drawback to animal production, as the low yields cannot adequately sustain animal production at a commercial level. Low yields may lead to very low minimal inhibitory concentrations (MICs), required to control enteric pathogens and promote the effective growth and health of the animal [83]. The increased demand for the plant’s essential oils from the food, beverages, cosmetics, and pharmaceutical industries has led to prices of the harvested plant and its extracts surging beyond levels sustainable for animal production. Harvesting is not monitored and regulated through sustainable and environmentally friendly policies, such as United Nations Convention on Biological Diversity and its Nagoya Protocol, leading to overexploitation and threatening the long-term continuity of the resource and population decline (Nott, 2019 https://bio-economy.org.za/resurrection-bush/ accessed on 15 March 2022). A solution to this would seem to be the use of greenhouses. Factors, such as temperature, disease (such as red spider mite infection), light intensities, soil composition and seasonal variation, and soil composition, need to be managed to improve the chemical composition of the plant in artificial conditions [84].

### 5.2. Bioaccessibility, Bioavailability, and Transfer Efficiencies of Phytogenic Nutrients

Past research focused on the biological and pharmacological properties of phytogenic compounds. These studies briefly describe the challenges of using herbal or natural plant additives in animal diets that need consideration to validate their health-promoting effects [85]. The main limiting factors towards the dietary inclusion of phytonutrients include low accessibility and bioavailability due to low solubility at neutral or acidic gut pH, conjugation, low stability, low absorption rates, quick passing, high rate of metabolism, and rapid elimination from the GIT [86]. Generally, the bioaccessibility and bioavailability of phytonutrients are low, as most exist as conjugates [87]. Bioaccessibility is described as the amount of food in absorbable form in the gut that has been released from the matrix by digestion. Bioavailability is defined as the portion of consumed feed, compound, or nutrient that is digested, absorbed, and utilised after transfer to the systemic circulation [88]. There is a lack of clarity and understanding regarding the quantification and modelling of the bioavailability of natural test compounds in in vitro assays and their predicted levels within in vivo systems [89]. As such, a lack in elucidating the specific molecular mechanisms of PFA actions in the biological system means that positive association between the intake of PFAs and perceived benefits, such as antioxidant defences, are not always guaranteed. It has been suggested that the effects of polyphenolics in vitro could be different in vivo, as evidence points to a difference between polyphenols metabolised and absorbed and those found in plants [90]. Extensive biotransformation in the small intestine and metabolic modification on absorption in the hepatic system converts dietary polyphenols from being highly potent in vitro into forms that have reduced potency in vivo [61]. The same authors further noted low absorption rates in flavonoid compounds from the gut and, effectively, their low concentrations in target cells, meaning there is little effect against free radicals and their products of metabolism. Further, radical molecular transformation of phytogenic compounds (up to 95%) through partial or full enzymatic hydrolysis leads to total changes in biological activities of resulting metabolites, thus, changing the expected rate and extent of absorption. The discrepancies between a high dietary concentration of bioactive phenols and available plasma concentrations of polyphenols in vivo have been noted. These concentrations could be lower than 1 µM in test subjects, meaning, therefore, not enough blood concentration levels to effect oxidant capacity [91]. Similarly, Alkhalidy et al. [87] observed very low plasma concentrations of phytochemical anthocyanins, approximately 30 nmol/L, after a high ingested dose of 50 mg, cleared a few hours after ingestion. The low plasma concentration of phytochemicals is due to rapid transformation of these compounds into conjugates, altering their structure along the GIT, thus, reducing their biological activities. As such, some suggestions that endogenously produced substances and dietary nutrients, such as albumin, vitamins C and E urate, and carotenoids, may be responsible for the antioxidant potential as compared to previously thought polyphenols [61,90]. Finally, 90–95% of polyphenols in the xenobiotics group undergo rapid metabolism and molecular changes once ingested and are immediately removed from circulation. In addition, some natural compounds may exist as esters and may undergo changes in in vivo systems and may, thus, be difficult to relate to specific biological activity.

### 5.3. Antinutritional Issues and Potential Adverse Effects and Toxicity

The notion that phytogenic plant material has been used in traditional medicine for centuries and is safe should not be taken for granted. Toxicology screening should precede any in vivo studies as these ingredient materials may contain potentially toxic properties [78]. As noted by Erhabor et al. [7], there is a dearth of literature on in vivo toxicology and toxicity and safety evaluation tests. Further, a lack of full elucidation of biological effects of phytogenic substances from *M. flabellifolius* is an impediment towards its use as an in-feed additive. Recent literature suggests that a high dose of phytogenic feed additives poses a threat to cellular processes and functions. Excessive doses of essential oils containing monoterpenes may promote pro-oxidant activity by damaging the mitochondria, disturbing electron flow, and producing reactive oxygen species [80,92]. Generally, flavonoids are antioxidants, but the presence of copper converts them to pro-oxidants. Similarly, over exposure of quercetin, highly present in *M. flabellifolius,* may be mutagenic and cause damage to DNA [87]. High-condensed tannins could bind biliary salts, a limiting factor for efficient fat digestion in poultry, leading to reduced absorption and increased excretion, and the inactivation of digestive enzymes [61]. In addition, high-condensed tannin doses inhibit mineral absorption, which may lead to consequences, such as anaemia. In addition, gastrointestinal lining damage and digestive enzyme destruction have also been reported [77].

## 6. Improving Bioavailability and Bioaccessibility of Nutrients in *Myrothamnus flabellifolius* (Welw.)

The nutritional and phytochemical composition of *M. flabellifolius* is affected by many factors, including season, biogeography, leaf, or shoot; in addition, post-harvest processes, such as preparation methods, processing temperature and time, or extraction method and protocol, have a large bearing on the product [16]. The extraction method depends on the bioactive compounds to be extracted and the plant tissue properties, temperature, time, particle size of the solid material, solvent system, and the ration of the solid to the extraction method.

### 6.1. Extraction of Essential Oil and Oleoresins

Recent research shows that the use of phytogenic plant products in solid ground/raw milled form in animal feeds is problematic due to issues, such as poor solubility, toxicity, antinutritional factors, and low biological activity. Essential oils are volatile lipophilic compounds derived through cold expression, alcohol, or steam distillation from leaf, stem, seed, and flower material, whilst oleoresins are nonaqueous-solvent-derived extracts [81]. Several factors affect the qualitative and quantitative composition of essential oil or extract from each individual plant species. These include location, solar radiation, season of harvest, temperature, genetics, moisture content, cultivar, drying conditions, and methods of isolation [93]. The main two distinct bio-synthetical groups of essential oils that are responsible for the biological properties are terpene hydrocarbons and aromatic oxygenated compounds. Yields of essential oil are, however, very low, expensive, and unsustainable to produce compared to extracts and synthetic products. Hydro-distillation of 10 species of Labiatae resulted in a yield of 0.75, 1.73, 0.30, 1.90, 0.13, 0.82, 0.52, 0.42, 0.25, and 1.99%, respectively [94]. *Myrothamus flabellifolius* hydro-distillation produced 0.04% wet weight of essential oil [8]. Three-hour steam distillation of freshly collected aerial parts weighing 6 kg yielded 0.73% and 0.67% (*w*/*w*) essential oil from *M. moschatus,* commonly found in Madagascar [6,11,72]. Chaturvedi et al. [11] found the following extract yields using different extraction methods: 14 (100% hexane), 32 (100% chloroform), 40 (100% methanol), 40 (70% methanol/water), and 48% (70% Ethanol/water) from sun-dried *M. flabellifolius,* collected in Botswana. A more recent study yielded 14.5% of extract using the boiling process on *M. flabellifolius* [16]. An essential oil yield of 0.04% wet weight was observed [8]. Similarly, Engelhardt et al. [17] found yields of 40 g AWEE, approximately 4% of the starting plant material. Chukwuma et al. [16] mined 14.74% extract from boiling powdered leaves in boiling water for 10 min and distilling crude extracts. In another study 1.015% of essential oil was derived using water distillation and absolute hexane [95]. The relatively low yields of obtention between 0.01 and 5% and the cost of infrastructure used for the oil extraction process is expensive and cannot be a method of choice for small-scale producers. The cost of production of essential oils can be offset by consumer demands, preference, and propensity to spend on animal products fed from clean, green, and ethically (CGE) sourced ingredients. Essential oils can, however, be included in animal diets at very low concentrations, approximately 500 mg/kg feed compared to 0.01–30 g/kg feed in raw form [81].

### 6.2. Microbial Fermentation

Plant material contains a diversity of compounds, including antinutritional factors, macronutrients, micronutrients, and phytochemicals. Upon processing, these chemicals interact and bind to each other to form insoluble complexes that entrap nutrients and reduce their bioavailability [96]. To improve the availability of nutrients after digestion, several methods are used to avail bound nutrients and unlock their pharmacological potential. The use of probiotic microorganisms to ferment feedstuffs has traditionally been used to improve gut microbial community, thus, promoting growth and feed efficiency in animals. Fermentation is a proven and feasible strategy that enhances the bio-functional properties of phytogenic compounds. This is achieved by microbial disruption of bonding in indigestible macromolecular structures of feed ingredients to release simple digestible essential nutrients accessible for animal use. This process simultaneously effects transformations of intricate structures into simpler molecules with reduced antinutritional factors and improved organoleptic characteristics (decreased bitterness and astringency). In addition, health-promoting prebiotic, probiotic, and symbiotic qualities increase the biological value of plant material after processing. Broilers fed a basal diet plus encapsulated microbial fermented herbal mix of Fermeherbafit (turmeric, ginger, garlic, noni, and *Moringa oleifera*) gained more weight than those fed a non-fermented herbal mix [85]. Further, Sugiharto et al. [97] observed improved erythrocyte counts, antibody titer towards NDV, LAB population, jejunal villi height, and blood lipid profile upon daily administration of fermented turmeric powder in diets. These results were corroborated by Qiao et al. [98], who observed improved average daily gains and feed conversion ratio in birds supplemented with fermented herbal material. The latter investigators suggested a link between the observed improvements to the ability of fermentation in unlocking the potential and improving the medicinal effects of herbs and other plant materials through biological modification. This modification reduces anti-nutritional factors, improves nutritional quality, increases phenolic content and digestible carbohydrate production, and lowers pathogenic microbes in the gut. In addition, the released simple carbohydrates, as well as polyphenols, are noted for their special ability to effect pharmacological properties, such as immunomodulatory, antitumour, antioxidant, hypoglycaemic, and anti-inflammatory effects. In addition, increased digestive enzyme secretion and intestinal nutrient digestion, and the modulation of gut microbiota with increasing proportion of beneficial microbiota were observed [99]. Furthermore, improved feed efficiency and intestinal function in birds supplemented with Aspergillus niger-fermented Ginkgo biloba leaves were linked to beneficial physiological mechanisms, such as increased protease and amylase activity and regulation of the sodium glucose co-transporter 1 (SGLT1) mRNA expression in the small intestine [100]. It should, however, be noted that the fermentation process requires a delicate balance of factors, such as temperature, time, and pressure to fully maximise nutrient harvest, lactic acetic, aethyl acetic and methylacetic acid production, carbohydrates, and polyphenol production. Hlahla et al. [101] increased total polyphenol content and reduced tannin content of *A. phylicoides* with an increase in higher fermentation temperatures. There is, however, a lack of science-based evaluation standards and systems and quality control protocol for the extraction of nutrients through microbial fermentation. In addition, the variations in quantity and potency of bioactive components present in *M. flabellifolius,* collected in different geographical locations and seasons, may lead to variation in outcomes of growth and immune response trials. Therefore, it is important to establish a standardized manufacturing practice and quality control protocols for phytogenic compound production and processing to ensure the quality of extracts [102].

### 6.3. Microencapsulation

Several bioactive compounds present in essential oils of phytogenic materials are very reactive, hydrophobic, sensitive to oxygen, light, temperature, and intrinsic and extrinsic pH [103]. They are prone to degradation in the upper gastrointestinal tract before they can be assimilated and used by the host animal, leading to low growth rates and poor economic results [81,97]. This limits their potential for use in animal diets, in addition to reduced natural flavours with increased undesirable and unpalatable sensory qualities, and decreased intake and ultimately low efficacy in animals [104]. As such, research on encapsulation technologies has gained much traction over the last decade due to several benefits observed, including protection of the bioactive components and allowing their release at a prescribed location and time [105]. Microencapsulation, a modern and novel technology very useful in animal nutrition, is described as finely dispersing or coating of a core material, usually a bioactive solid, liquid, or gaseous component, antioxidants, enzymes, cells, microorganisms, or any other substance with a shell/matrix/protective wall material, such as protein, lipids, alginates, gums, or carbohydrate, resulting in a smaller millimetric capsule [106,107]. The platform’s advantages are based upon the high specific-area-to-volume and porous nature, which allows nutrient infiltration, stable structure, and high drug-release rates [108]. In the process, encapsulation enhances the bioavailability of sensitive compounds [109] and masks the unpalatable taste and odour of some ingredients [110]. Surai [61] notes that polyphenols have a higher beneficial impact through antioxidant protection if they are in contact with gut-lining cells, provided they do not undergo digestion and absorption in the upper gastrointestinal tract, highlighting the need for encapsulated protection of additives. Comparing the effect of supplementing powdered (menthol and anethole at 150 mg/kg) vs. matrix-encapsulated (carvacrol, thymol, and limonene at 100 mg/kg) phytogenic feed additives, Hafeez et al. [111] observed improved feed conversion efficiency, stimulation of digestive enzymes, and apparent ileal digestibility of nutrients in broilers. A novel encapsulated turmeric nanofiber phytogenic feed additive formulation proved to be a potential replacement for the antibiotic growth promoter by increasing immunity resistance, ileal crypts depths, faecal dry matter content (which reduced cholesterol levels), and promoting eubiosis in broiler chickens [109]. The putative mechanism involves the presence of triterpenoids in the active components of PFA, which are protected in the stomach but slowly released from the capsule matrix at high doses into the specific segment of the lower gastrointestinal tract, favouring its action [112]. In the lower GIT, the active compounds protect the cells and tissues against free radicals from cellular redox processes as well as against inflammation brought about by cytokines [80]. These authors also observed weight gain in lambs supplemented with a micro-encapsulated blend of carvacrol, thymol, and cinnamaldehyde and attributed it to the microencapsulated herbal component passing through rumen and is only released in the intestine.

### 6.4. Blending of M. flabellifolius with Complimentary Compounds

Phytogenic feed additives contain a large variety of compounds and active ingredients and, therefore, a different chemical profile exists, even within each group of plant species due to factors described earlier. In addition, processing and manufacturing, storage, volatility and the reactive nature of extract, effective dose, and animal environmental and health status may not guarantee a certain concentration of the bioactive ingredient [57]. As such, some bioactive compounds expected to be present may be lacking in the final additive upon feeding, resulting in unexpected, non-effectiveness in animals and inconsistency in obtained results. Moreover, strong odours of some PFAs may pose a challenge to intake, as well as evidence suggesting that without protection, some phytogenic active substances are easily absorbed in the upper GIT. As such, research to guarantee minimum inhibitory concentration (MICs) and efficacy in growth performance has led to observations of positive beneficial results in the use of blending phytogenic additives with other compounds. Research on the synergistic effects of blending phytogenic substances with other compounds to find alternatives to growth-promoting antibiotics and other ionophore coccidiostats has, thus, received considerable attention over the last few years. Most of the published studies demonstrated the beneficial complimentary effect shown through enhanced efficacy on growth performance, gut microflora, and intestinal health when phytogenic additives were used in combination with other compounds, rather than as individuals [113]. Phytogenic feed additives can be blended with other phytogenic extracts, essential oils, or organic acids. The blending combination advantages include (i) increase in spectrum of activity, (ii) reduction in toxic or adverse side effects, (iii) decrease in required doses and costs, (iv) elimination of drug resistance, and (v) improvement in additive and synergistic activity, as explained below [114]. Among these alternatives, the most promising method uses a combination of hydrophobic phytogenic essential oils with lipophilic organic acids, taking advantage of their different physiological effects and complementary modes of action and, thus, exerting a synergistic and additive beneficial effect on animal growth performance and gut health [57]. Effective and weak organic acids with a lower pKa value (pH at which 50% is dissociated and non-dissociated; 2.93–4.88) including short-chain fatty acids, e.g., acetic, malic, citric, formic, propionic, and butyric acids, are more commonly used in animal health and nutrition as feed acidifiers due to the broad spectrum of activities and synergistic effects of different pKa values [115]. The mode of action, as described by Abdelli et al. [57], suggests the disruptive nature of hydrophobic essential oils on Gram-positive bacterial cell walls and membranes, allowing access of organic acids and other phenolic compounds into the cytoplasm, where they disturb the proton and anion concentrations and inhibit enzymatic pathways. This subsequently renders the cell porous to leakage of other vital cellular components, acidifying the cell interior, leading to subsequent damage and cell death (81,114,115). Another suggestion was that combination increases the number size and duration of the number of channel pores in the membrane, which would allow more bacterial cell damage [116]. Liu et al. [117] and El-Azrak et al. [118] described how residue build-up, side effects, and antibiotic resistance were unlikely with the use of dietary phytogenic additives and complimentary synergistic blends of essential oils composed of phenylpropanoids and monoterpene hydrocarbons than when offered individually. Hafeez et al. [111] weighed in on the importance of terpene composition in enhancing the efficacy of phytogenic additives as well as endogenous secretions. Patel and Goyal [44] reported greater additive effects on pancreatic enzyme stimulation with mixtures of phytogenic feed material than when fed individually. In a study to evaluate the effect of phytogenic additives and organic acids in combination or alone, there was no effect on growth performance but enhanced intestinal quality and immune responses were observed [119].

An effective blend of phytogenic additive ingredients should exhibit minimum or no antagonistic effects on animal physiological and metabolic processes but should fully exploit the additive and complementary effects that modulate feed intake and body weight changes in livestock. It is important to elucidate the chemical profile of each potential candidate ingredient and understand the mode of action of each bioactive ingredient before blending with *M. flabellifolius* to improve efficacy and potency towards animal growth and health. This is important to achieve a standardized optimum dose of bioactive phytogenic ingredients and their combinations in relation to different animal species and physiological status [120] and prevent antagonistic effects or, in the worst cases, poisoning. Possible combinations/blends should be able to guarantee high monoterpene and phenylpropanoid content, which increase antioxidant and antimicrobial capacity in the additive and resulting feed, characteristics that are directly related to improved growth and health of livestock animals. It is, however, important to acknowledge mechanisms of an antagonistic nature that are currently still speculative, as most research attributes poor growth rates to dose effects, health challenges, and management practices. Lastly, prices of ingredients and economic analysis of phytogenic studies indicate that they may not be feasible and economical to generate maximum profit for the business enterprise [103]. It is also worth noting that blending reduces the costs associated with phytogenic feed additives, by reducing the quantity needed to reach minimum inhibitory concentrations (MICs) of other expensive ingredients [57]. Hassan et al. [121] found higher net profit per kilogram liveweight in broiler birds supplemented with a combination of artichoke extract and organic acid than in non-supplemented due to feed utilisation.

## 7. Future Perspective and Conclusions

Observations on current research indicate that the use of phytogenic feed additives, such as *M. flabellifolius,* holds the most promising solution towards the replacement of in-feed antibiotics due to the presence of various secondary metabolites. These secondary metabolites impart antimicrobial, antiviral, antioxidant, and gut improvement properties on the animal, which help improve growth performance. However, several limitations still exist for the uptake and commercial use of *M. flabellifolius*. Research towards quantifying secondary metabolites, systematic approaches to understanding their efficacy, and the mode of action of each active compound, as well as possible synergistic and antagonistic interaction with other phytogenic feed additives and feed, is still necessary, In addition, economic and environmental challenges brought about by both the methods in use and conventional organic solvents used in the extraction of secondary metabolites, which may not conform to green/sustainable chemistry, need to be addressed. In conclusion, the review managed to gather the latest literature regarding the use of *M. flabellifolius,* in both human and animal studies, and related it to its potential use in animal nutrition. The study managed to describe key nutritional and non-nutritional secondary metabolites present in the plant, their mode of action, the limitations towards its use, and technologies, such as blending, microbial fermentation, encapsulation, and essential oil extraction, that can be used to enhance or improve its use as an alternative replacement to antibiotic growth promoters. Moreover, effects on growth performance, gut function, and animal health have been described in detail, as well as issues surrounding safety and toxicity. Finally, the use of *M. flabellifolius* in animal diets is a novel and promising approach towards the elimination of antibiotics in feed and animal production.

## Figures and Tables

**Figure 1 animals-12-01973-f001:**
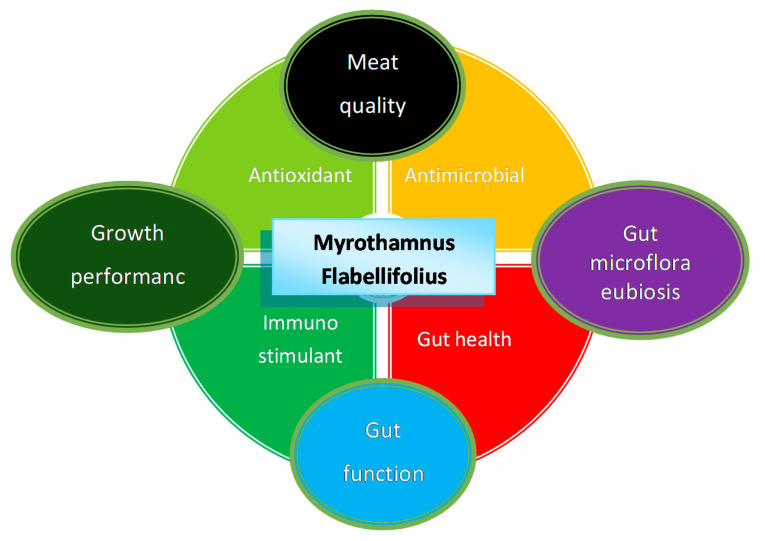
Mode of action based on secondary metabolites present in *M. flabellifolius*. The relationship between antioxidant, antimicrobial, immunostimulant, and gut health improvement properties of *M. flabellifolius* and how they affect growth performance, gut microflora, gut function, and meat quality.

**Table 1 animals-12-01973-t001:** Phytochemicals present in *Myrothamnus flabellifolius* and potential benefits.

	Quantity	Potential Mode of Action	Reference
Nutrient			
Trehalose	34.7 ± 7.2 mgg^−1^	proteins and lipid stability	[25]
Raffinose	2.49 g/100 g	osmotic stress protection	[16]
Gallic acid	26.62%	Anti-obesity	[16]
Ferulic acid	15.23%	Antiinflammation	[16]
Stachyose	2.18 g/100 g	osmotic stress protection	[16]
Sucrose	56.5 ± 6.6 mgg^−1^		[25]
Phytochemical			
Carvacrol	-	Antibacterial, antioxidant	[17,57]
3,4,5 tri-O-galloylquicic acid	0.73, 0.32, and 0.0029 g/100 g	Membrane protectant, Anti-viral reverse transcriptase activity,	[3,4]
Galloylquinic acid	-	Antioxidant	[58]
Arbutin	-	Melanogenesis inhibitor	[23]
Ellagic acid	-	Antioxidant	[59]
Gallocatechin	1.43 ± 0.03 mg/g	Chemoprotective, antioxidant, antibacterial	[4,56]
Trans-pinocarveol	19.57%	Antimicrobial	[8]
Quercetin glucoside	3 mg AWE_w_	Antioxidant	[60]
Pinocarvone	11.13%	Antimicrobial	[8]

## Data Availability

Not applicable.

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
