# Peer review of "Exploring the Potential of Myrothamnus flabellifolius Welw. (Resurrection Tree) as a Phytogenic Feed Additive in Animal Nutrition"

_animals, 2022, doi:10.3390/ani12151973_

Round 1

Reviewer 1 Report

The authors provided an interesting review and of importance to the field based on current trends in animal nutrition. However, the text is lengthy not well summarized, and very repetitive.  This increases the challenge to provide an in-depth review. I strongly suggest the use of tables and graphs to consolidate the data and facilitate the overall flow of the manuscript.

Additionally, the authors are not referencing correctly in the text. When authors refer to a specific manuscript in the text then you have to describe the authors e.g. These results were supported by  Zhang et al [30] where they fed .....

Not only the number.  This issue also difficult revision and loses the flow of the paper to know when are you changing ideas, authors, etc. 

Finally, the authors are strongly encouraged to remove any repetitive ideas throughout the text that are only adding more text and not content. 

More comments in the attached pdf

Author Response

Dear Reviewer. Thank you for the insightful review of our manuscript. Attached below are the corrections made to the manuscript as per your suggestions. In the revised manuscript we have marked the corrected parts with track changes mode.  

The authors provided an interesting review and of importance to the field based on current trends in animal nutrition.

We appreciate the positive feedback. 

However, the text is lengthy not well summarized, and very repetitive.  This increases the challenge to provide an in-depth review.

We thank you for this insightful suggestion. We have managed to trim some of the literature and removed all repeated literature as suggested. For example we removed section describing the Free radical species metabolism in section 4.1. In Section 5.2 we removed some sections which we thought were not necessary for the manuscript. To summarise the literature throughout the document, we removed that words, phrases and statements that added length but did not improve the flow of ideas in the manuscript. 

I strongly suggest the use of tables and graphs to consolidate the data and facilitate the overall flow of the manuscript.

We appreciate this suggestion. We managed to improve Table 1 in adding to the description of the chemical composition of M. Flabellifolius as well as add quantities of each compound and perceived benefits. We were not however able to add more tables and figures to the manuscript due to various limitations beyond our control but feel that we have added appreciable literature.

Additionally, the authors are not referencing correctly in the text. When authors refer to a specific manuscript in the text then you have to describe the authors e.g. These results were supported by  Zhang et al [30] where they fed .....

Thank you for this suggestion. Managed to go through the document and corrected such referencing mistakes which started at reference number 22 Colaco and Roser et al line 126 and throughout the manuscript. We managed to align the referencing style to that of the journal as shown in the attached document.

Not only the number.  This issue also difficult revision and loses the flow of the paper to know when are you changing ideas, authors, etc. 

We appreciate this comment and suggestion. We managed to revise the manuscript with the aid of an English editor to improve the flow of ideas. Several long statements were shortened, appropriate grammar was used throughout the document to link ideas. Ideas in different paragraphs were linked using proper scientific terminology to 

Finally, the authors are strongly encouraged to remove any repetitive ideas throughout the text that are only adding more text and not content.

The authors managed to read and correct these mistakes. Several changes were throughout the document, removing unnecessary and unrelated issues in order to improve the quality of the manuscript. For example  literature describing the dearth on information relating to Myrothamnus Flabellifolius research were removed from several sections of the manuscripts. Also, literature describing its uses and competition related to demand was removed in order to focus the manuscript and make it a better read. 

More comments in the attached pdf

We appreciate and thank you for the attached comments. We managed to address all these mistakes as suggested by the reviewer to make the manuscript a better read. Please find the attached improved manuscript in track changes mode.

Reviewer 2 Report

The manuscript submitted by Tawanda Nantapo and Marume, is focused on the use of Myrothamnus flabellifolius. as a phytogenic feed additive in animal nutrition.

The topic is quite interesting and the paper is well written and properly organized. Only minor aspects need to be addressed before further considering the paper for publication.

Specific aspects:

-          L24: “has been traditionally used…”.

-          L60: “have been widely demonstrated…”.

-          L91-92: I invite the authors to double check the reported percentages. The mineral content seems very high and, moreover, adding the percentages of fat, proteins, fibers, carbohydrates and minerals we are abundantly above 100%.

-          L329: It would be really useful in my opinion if the authors summarized the potential benefits of M. flabellifolius in animal production and health by preparing a table indicating the most pertinent bibliographic references. I believe these are fundamental steps to improve the communicative effectiveness of a review paper.

-          Figure 1 is quite informative, however I suggest implementing the description of the image in the caption.

-          L340: The starting sentence of the paragraph should be checked. There was probably a pagination problem.

-          The references style must be reviased according to the Journal recommendations.

Reviewer 3 Report

This is an interesting review-paper discussing the possible use of Myrothamnus flabellifolius as natural feed additive in animal nutrition.

The topic fits well the overall scope of Animals journal and the topic investigated is of interest for animal nutrition and health.

The paper is properly organized and the different sections result well structured.

However, some changes may add further value to the manuscript, such as:

- An overall check of the English language is suggested; some sentences are quite long and thus hard to follow for a reader;

- Some of the titles of the different paragraphs could be shortened;

- A figure representing the Myrothamnus flabellifolius tree could be useful for the readers;

- Check if all the references have been cited into the text or in the references' list;

- the sectio "7. Future perspective and conclusions" could be further improved.

Author Response

Reviewer 3:

This is an interesting review-paper discussing the possible use of Myrothamnus flabellifolius as natural feed additive in animal nutrition. 

Thank you very much for the positive feedback

The topic fits well the overall scope of Animals journal and the topic investigated is of interest for animal nutrition and health.

Thank you very much for the positive comment

The paper is properly organized and the different sections result well structured.

Thank you very much for this positive feedback

However, some changes may add further value to the manuscript, such as:

  • An overall check of the English language is suggested; some sentences are quite long and thus hard to follow for a reader;

Thank you very much for the comment. We have gone through the paper several times with the help of an English editor to correct some of these grammatical errors and we have noted and shorted several sentences within text as you have suggested to make it a better and easy read in relaying our findings

  • Some of the titles of the different paragraphs could be shortened;

The following titles were shortened

Section 3.0 title "Phytochemicals present in Myrothamnus flabellifolius (Welw.) and its extracts" was shortened to, "Phytochemicals composition of Myrothamnus flabellifolius (Welw.)".

Section 4.0 title  "Overview of potential benefits of M. flabellifolius in animal production and health" was shortened to "Potential animal production and health benefits"

4.2 "Antimicrobial and immune inhibition of pathogenic microbes", shortened to "Antimicrobial characteristics".

4.3 "Feed quality, growth performance, gut function, gut health and animal product quality", shortened to "Growth performance and health effects".

  • A figure representing the Myrothamnus flabellifolius tree could be useful for the readers.

A figure representing Myrothamnus flabellifolius tree could not be added to the manuscript at the moment as we experience technical issues. A laptop with images crashed and also field sites could not be reached due to logistical challenges. We hope in future we will request the Editor to attach images as an appendix when they are available.

Check if all the references have been cited into the text or in the references' list;

All references were cross-checked in text and in the reference list using manual and automatic Mendeley referencing application.

  • the section "7. Future perspective and conclusions" could be further improved

Thank you very much for the recommendation. We have edited and modified the content in this section to reflect on the title.

Round 2

Reviewer 1 Report

Good work addressing comments